# WFD Ecological Quality Indicators Are Poorly Correlated with Water Levels in River Catchments in Tuscany (Italy)

Chiara Arrighi [1,*], Isabella Bonamini [2], Cristina Simoncini [2], Stefano Bartalesi [2] and Fabio Castelli [1]

1   Department of Civil and Environmental Engineering, Università degli Studi di Firenze, 50139 Florence, Italy; fabio.castelli@unifi.it
2   Water Resources Office, Northern Apennines River Basin District, 50122 Florence, Italy; i.bonamini@appenninosettentrionale.it (I.B.); c.simoncini@appenninosettentrionale.it (C.S.); s.bartalesi@appenninosettentrionale.it (S.B.)
*   Correspondence: chiara.arrighi@unifi.it

**Abstract:** The Water Framework Directive (WFD) requires European countries to achieve a good ecological status of surface water bodies and demands that River District Authorities define ecological flows consistently. Nevertheless, the relationship between ecological and hydrological indicators is not fully understood and not straightforward to apply because ecological and hydrological indicators are monitored by different institutions, with different timings and purposes. This work examines the correlation between a set of ecological indicators monitored by environmental agencies (STAR-ICMi, LIMeco, IBMR, and TDI) and water levels with assigned durations monitored by the hydrologic service in Tuscany (central Italy). Reference water levels are derived from stage-duration curves obtained by the statistical analysis of daily levels in the same year of ecological sampling. The two datasets are paired through a geospatial association for the same river reach and the correlation is measured through Pearson's r. The results show poor correlation (r between −0.33 and −0.42) between ecological indicators and hydrologic variables, confirming the findings observed in other Italian catchments with different hydrologic regimes, climate, and anthropogenic pressures. Nevertheless, the negative correlations show a decreasing water quality with water depths, i.e., in the lower part of the catchments more affected by anthropogenic pressures. These findings suggests that the determination of ecological flows with a purely hydrological approach is not sufficient for achieving WFD objectives in the study area.

**Keywords:** ecological flow; surface water hydrology; water resources management





## 1. Introduction

The conservation of aquatic ecosystems in terms of quantity and quality is one of the sustainability challenges faced by many countries that exploit water resources for energy, agricultural, industrial, and domestic uses [1]. The achievement of good ecological status in natural surface water bodies is a key objective of the European Water Framework Directive (WFD) [2,3]. The WFD recognizes the critical role of water quantity and dynamics in supporting the quality of aquatic ecosystems and requires River Basin District Authorities to set out ecological flows for the river network as a part of the Water Management Plan. Ecological flows are considered within the context of the WFD as "an hydrological regime consistent with the achievement of the environmental objectives of the WFD in natural surface water bodies" [2].

The ecological integrity of water bodies is measured by means of biological indicators, such as macroinvertebrate based indices, which are sensitive to pollution and habitat degradation [4–6]. The quantification of the flow–ecology relationship is fundamental for defining ecological flows [7–9]; however, although several works addressed the problem for regulated rivers and hydropower abstractions [10–15], the correlation between ecological and hydrological parameters is still debated. Most of the studies adopt habitat models

designed to establishing weighted usable area for selected species and require the knowledge of water depths and velocity at river cross sections and are applicable to single river reaches [16,17]. The relationship between ecological status indicators and flow parameters in a large Alpine catchment was recently studied and highlighted a poor correlation [18]. Particularly, the study examined the STAR_ICMi index (STAndardisation of River classifications Intercalibration Common Metric index) [19], which is a macroinvertebrate-based index, officially adopted by the Italian legislation to classify the ecological status of river waters in the WFD framework. The index is also adopted in other European Union (EU) countries and merges several metrics, including taxonomic richness and diversity. The index showed a low sensitivity to discharge in Alpine and perialpine rivers, which usually are affected by hydropower alterations but have low pollution [5,19–23].

Among the ecological indicators used to assess the ecological status of water bodies, there is the LIMeco index (macro-descriptor pollution level for ecological status) [24], which classifies river water quality in terms of dissolved oxygen and nutrient concentration (nitrogen and phosphorus). Although the LIMeco index is more a chemical indicator than an ecological indicator, it is widely used as a proxy for ecological quality. Other indices consider the presence of macrophytes (Macrophyte Biological Index for Rivers, IBMR) and diatom (Trophic Diatom Index, TDI) [25–27]. The integrity and continuity of river morphology and riparian zones is also included in the river quality classification sensu WFD [28,29].

In addition to the difficulties in understanding the response of ecological indicators to flow alterations, a major problem is the identification of ecological flows at large scales, i.e., district scale, as required by the Water Management Plans [30,31]. Hydrological methods identifying natural flow regimes [32–34] are often used for the assessment of ecological flows. In Romania, a hydrological method combined with monthly hydrological forecast was developed to compute ecological flow [34]. Machine learning methods were also applied to predict low flows [35]. Flow intermittency and drought effects on environmental flows have also been examined [17,36].

In the context of supporting the definition of ecological flows at the regional scale, this work aims at preliminarily analyzing the correlation between official ecological indicators (sensu WFD), i.e., STAR_ICMi, LIMeco, diatom (TDI), and macrophytes (IBRM) and characteristics water levels derived from stage-duration curves for the river catchments in Tuscany (central Italy), which have climatic conditions and anthropogenic pressures quite different from those examined in previous work. The use of geographic information systems (GIS) allows for the pairing of ecological and hydrological indicators that are monitored by different institutions and have different timing and purposes, and thus are not immediately comparable.

## 2. Materials and Methods

### 2.1. Study Area

Tuscany is a region in central Italy with a surface area of c. 23,000 km$^2$ and a population of approximately 3.7 million. The Arno River has the largest river basin in the region (Figure 1) with a surface of 8200 km$^2$. Other important river basins are the Ombrone (4700 km$^2$), Serchio (1500 km$^2$), Magra (1700 km$^2$), and Fiora (825 km$^2$). The northern part of the territory is characterized by mountains, with an altitude of the order of 1000–1500 m a.s.l., the western part is bounded by the Tyrrenian sea. Climatic conditions are semiarid in southern coastal areas and perhumid in the northern mountainous regions.

Ecological indicators are monitored by the Environmental Protection Agency of Tuscany (ARPAT) according to the institutional program for the achievement of WFD objectives and are available in a public database [37]. Although samples of the components of the ecological indicators are collected 1–3 times a year, the aggregated indices, such as the LIMeco or STAR_ICMi, are averaged to provide one value per year. Ecological monitoring is less frequent for water bodies with high quality and the achievement of WFD objectives is determined every three years. In the current triennium 2019–2021, data for 2021 are still

under elaboration, thus the data of 2019–2020 are selected for this preliminary analysis (c. 300 monitored sites). The regional hydrologic service (SIR) provides daily water levels at the hydrometric gauges for 172 stations. The hydrometric data of the years 2019–2020 were selected consistently with ecological data.

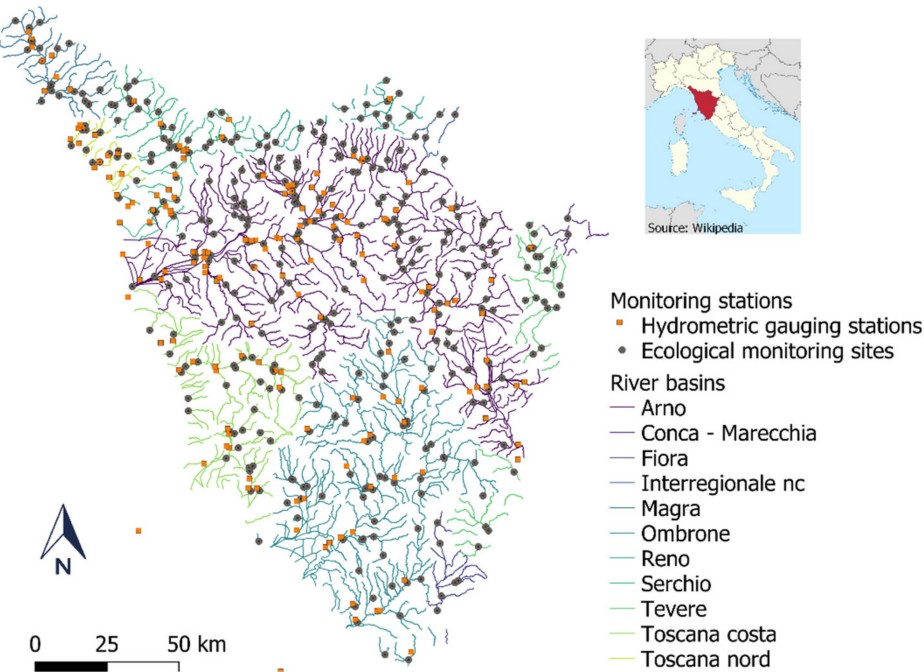

**Figure 1.** Map of the river network of the catchments in Tuscany, hydrometric gauging stations (SIR), and ecological monitoring sites (ARPAT).

*2.2. Methods*

Stage-duration curves are derived for the same year of the ecological status classification made by ARPAT to obtain the characteristic water levels that occur with a certain frequency in river reaches. A stage-duration curve is similar to a flow-duration curve, i.e., a cumulative frequency curve that shows the percent of time or days specified discharges were equaled or exceeded during a given period, but it refers to measured water levels instead of discharges [38]. The stage-duration curve combines the water level characteristics of a stream throughout the annual range, without regard to the sequence of occurrence (top-right of Figure 2). Duration curves have a long history in the field of water-resource engineering and are also widely applied for the determination of minimum vital flow [2,39]. In this work, six reference water levels $H_i$, where $i$ is the number of days the water level is equaled or exceeded, are derived from the stage-duration curves and used for correlation analysis. They are $H_1$ (annual maximum), $H_7$, $H_{274}$, $H_{358}$ (formerly used to determine minimum vital flow in Arno river basin), $H_{365}$ (annual minimum), and the difference between maximum and minimum levels $H_1$–$H_{365}$. The selection of these reference water levels glances at the flow parameters identified as indicators of hydrologic alteration (IHA) [12,40].

Because the hydrological and ecological status datasets are collected by two different institutions, they do not have common attributes to be associated but have geographic coordinates. Thus, they are combined by using a geospatial association in a GIS environment before the correlation analysis. Each single river reach, i.e., the segment between two confluences, is assigned a unique ID with the split with lines function in QGIS. The two points datasets (Figure 2) are then assigned the river reach ID with the join attribute by location function. Finally, the attributes of ecological monitoring site and hydrometric gauging station are joined through the river ID. This procedure avoids an incorrect associa-

tion by distance that could confuse two near points belonging to different reaches (bottom of Figure 2).

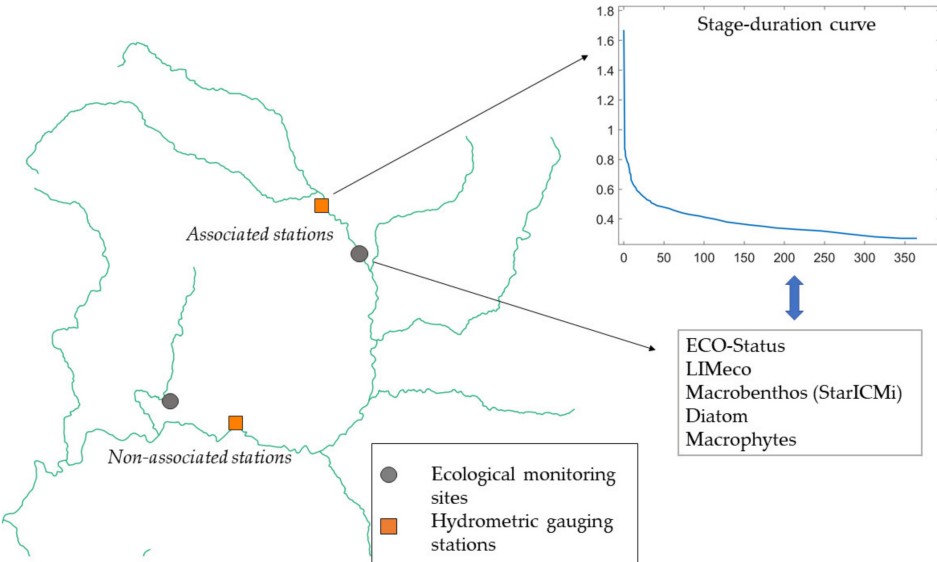

**Figure 2.** Example of GIS-based association of hydrometric gauging stations (SIR) and ecological monitoring sites (ARPAT). The top right panel shows the stage-duration curve with levels (Y-axis) expressed in m and duration in days (X-axis). The bottom right box lists the ecological indicators paired with each reference water level $H_i$.

The correlation analysis is performed by calculating Pearson's correlation coefficient *r*, which measures the linear correlation between two sets of random variables *A, B:*

$$r(A, B) = \frac{cov(A, B)}{\sigma_A \sigma_B} \qquad (1)$$

where *cov* (*A, B*) is the covariance of *A* and *B*, and $\sigma_A$, $\sigma_B$ are the standard deviations of *A* and *B*, respectively. The coefficient *r* is between −1 and 1. In MATLAB, the Pearson's *r* is calculated with the function *corr* (argument, 'Type', 'Pearson').

### 3. Results

The association between hydrometric gauging stations and ecological monitoring sites yields 67 couples of points to be used for the correlation analysis (Table S1, Supplementary Materials). The limited number of associated points is a consequence of the different purposes for which the hydrometric and ecological monitoring systems are set up. In fact, most ecological monitoring sites are located in low order streams, i.e., in upper catchments or in flat areas hosting wetlands, whereas hydrometric stations are placed in the lower part of the catchments with the purpose of flood warning. Figure 3 shows a summary plot of the ecological indicators and the characteristic water levels. In addition to the ecological indices, such as the LIMeco index, the overall ecological status, which is the final classification attributed to the water body, is also plotted against water levels. The ecological status ranges from 1 (very high) to 5 (poor). Table 1 presents the Pearson's correlation coefficient matrix. As can be seen in Figure 3 and Table 1, the correlation between ecological indicators and reference water levels is quite low. The statistical significance of the correlations was evaluated with a *t*-test, which rejected the null hypothesis at the 5% significance level.

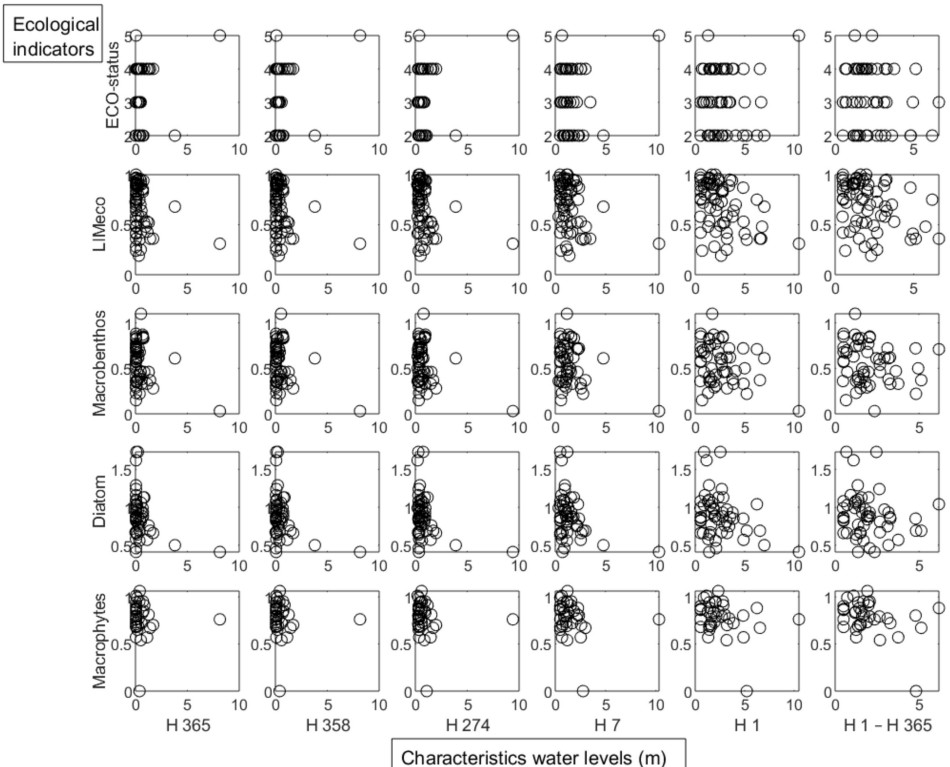

**Figure 3.** Scatter plots of ecological indicators versus characteristic water levels.

**Table 1.** Pearson's correlation coefficient r for all ecological and hydrological indicators. Blue cells show higher correlation values between ecological and hydrological indicators, orange cells show the higher correlations among ecological indicators. Abbreviations in the table: macrobenthos (MB), macrophytes (MP), ecological status (ECS).

| | ECS | LIMeco | MB | Diatom | MP | $H_{365}$ | $H_{358}$ | $H_{274}$ | $H_7$ | $H_1$ | $H_1$–$H_{365}$ |
|---|---|---|---|---|---|---|---|---|---|---|---|
| ECO-Status | 1 | | | | | | | | | | |
| LIMeco | −0.47 | 1 | | | | | | | | | |
| Macrobenthos | −0.80 | 0.62 | 1 | | | | | | | | |
| Diatom | −0.32 | 0.34 | 0.43 | 1 | | | | | | | |
| Macrophytes | −0.19 | 0.58 | 0.56 | 0.46 | 1 | | | | | | |
| $H_{365}$ | 0.26 | −0.30 | −0.32 | −0.34 | −0.07 | 1 | | | | | |
| $H_{358}$ | 0.25 | −0.30 | −0.31 | −0.33 | −0.07 | 1 | 1 | | | | |
| $H_{274}$ | 0.25 | −0.33 | −0.33 | −0.33 | −0.10 | 0.99 | 0.99 | 1 | | | |
| $H_7$ | 0.18 | −0.35 | −0.32 | −0.35 | −0.18 | 0.91 | 0.91 | 0.94 | 1 | | |
| $H_1$ | 0.09 | −0.42 | −0.32 | −0.33 | −0.29 | 0.67 | 0.68 | 0.70 | 0.87 | 1 | |
| $H_1$–$H_{365}$ | −0.09 | −0.34 | −0.17 | −0.17 | −0.37 | 0.14 | 0.15 | 0.19 | 0.47 | 0.83 | 1 |

In the right part of the table, high values denote the high correlation among water levels derived for assigned durations. In the lower- and upper-left part of the table, the correlation between ecological quality indicators and water levels and among quality indicators, respectively, are shown.

The LIMeco index shows a non-negligible correlation with macrophytes (r = 0.58) and macrobenthos (r = 0.62) indicators. Macrobenthos and macrophytes are also correlated (r = 0.56). With respect to water levels, correlations are lower, although statistically significant according to the *t*-test. The LIMeco index has r = −0.42 with the annual maximum discharge $H_1$ and r in the range −0.33–0.35 with $H_{274}$, $H_7$ and $H_1$–$H_{365}$. The correlation has a negative sign because ecological indicators decrease with increasing water levels.

Macrophytes show some limited correlation with $H_1$–$H_{365}$ (r = −0.37) that can be interpreted as the importance that water level fluctuations have for the development of aquatic plants in the riparian zones.

## 4. Discussion

The preliminary results obtained for the river catchments in Tuscany, which have different climatic, morphological, and socio-economic conditions with respect to other study areas [18,21], confirm a poor correlation between ecological indicators and hydrologic variables. However, the negative value of Pearson's r shows that, overall, ecological quality indicators tend to decrease with increasing water levels, meaning that lower parts of the catchments have a deteriorated water quality. Low correlation among ecological indicators confirms that this selection is sound because they are not redundant in describing surface water quality aspects.

Criticisms of the use of ecological indicators, e.g., Macrobenthos (STAR_ICMi) index to determine ecological flows [20,23], are thus fully understandable, especially when the main pressures on surface water bodies are related to flow regulation, e.g., hydropower. The ecological indicators here analyzed, which are the pillars of the institutional monitoring of environmental protection agencies in Italy, were selected to detect water quality alterations more than quantity ones. In fact, the LIMeco index, which is more correlated to characteristic water levels, can be seen as a proxy of the river capability to dilute nutrients and organic load; it is indeed more a chemical indicator sensitive to river discharges than an ecological indicator. The LIMeco index is nevertheless one of the less sensitive indicators in the study area according to ARPAT reports and should not be considered alone as a reference for ecological flows determination [37]. In the study area, the diatom indicator (TDI) was also found to be poorly sensitive in the monitored sites, whereas the macrophytes IBMR and STAR_ICMi were found to be very sensitive ones, with the highest number of observed poor and very poor statuses [37]. IBMR is based on field cover percentage, the species trophic score, and a coefficient of ecological amplitude and was found to be sensitive to discharge diversion in rivers also without pollution, in this sense the correlation (r = 0.37) found with $H_1$–$H_{365}$, i.e., the water level range, supports this observation. Moreover, hydromorphological aspects are not fully taken into consideration in the STAR_ICMi index, and other indicators, such as the Lentic-lotic River Descriptor (LRD), could be potentially more significant in describing invertebrate communities [41].

The limited number of paired ecological-hydrological data can be not representative of the whole catchment portions. In fact, low order streams, e.g., mountain river reaches, or wetlands are not covered by the hydrometric gauging stations. This reduces the examined sample to middle and low catchment areas and might cause a bias in the statistical analysis. Moreover, coupling hydrometric gauging stations and ecological monitoring sites that do not geographically coincide might introduce uncertainties in paired water levels, which can differ significantly among river cross sections.

Although the study area is subject to significant anthropogenic pressures other than hydropower regulation, such as intensive agriculture and civil/industrial wastewater treatment plants, the poor correlation among ecological indicators and water levels suggests that pure hydrologic approaches based on natural flows [32] in the determination of ecological flows are not sufficient. Natural flows, i.e., flow that would naturally occur in absence of diversions, regulations, or withdrawals, are probably too low to balance quality alterations in catchments affected by high anthropization and to achieve the WFD objectives. Nevertheless, hydrological models have the advantage of being less expensive than hydraulic-habitat and holistic methodologies; they are also faster and applicable for regional assessment [2]. In fact, hydrological methods are still widely applied for the assessment of natural flows [31,33,34] and for the understanding of drought consequences on river flows [36]. The methods widely applied to the river reach scale, which require on-site inspection, data collection, and modelling of habitat suitability, are more difficult to adopt for regional scale water resources planning [42,43].

Climate change is also expected to decrease low flows in the Mediterranean region [44,45], where negative trends have already been observed in low flows [46]. A better understanding of climate change effects on ecological flows and quality indicators in areas subject to different climatic and pressure conditions should be sought. Further research

could explore the relationship between ecological indicators and pressure indicators, such as the percentage of agricultural/urban/forest areas in the contributing catchment, river morphological alterations, or wastewater volumes released. Other indicators to be examined could be purely climatological, such as seasonal precipitation and mean temperature in the contributing catchment, also in a climate change context. Machine learning methods could be potentially helpful in the presence of a sufficient number of training and validation data, because large scale hydrological models capable of accounting for ecological processes and or transport of nutrients would be quite complex to set up and validate [47]. Machine learning methods were recently applied for quality modelling with promising results [48–50].

The increase in paired ecological and hydrological indicators could be achieved by means of regional hydrological models, which could provide discharge-duration curves in correspondence of all ecological monitoring sites.

From a researcher perspective, a denser monitoring network of both ecological and hydrological variables, also in the same river cross section, would be a valuable way to improve the understanding of flow-ecology relationship and support water resources planning in the framework of WFD and in the uncertainties around spatial and temporal patterns of rainfall and extreme temperatures.

**Supplementary Materials:** The following are available online at https://www.mdpi.com/article/10.3390/hydrology8040185/s1, Table S1: Data of the paired ecological and hydrological monitoring sites.

**Author Contributions:** Conceptualization, methodology, writing—original draft preparation, C.A.; data selection C.A., I.B., C.S. and S.B.; supervision, F.C. All authors have read and agreed to the published version of the manuscript.

**Funding:** This research didn't receive any funding.

**Institutional Review Board Statement:** Not applicable.

**Informed Consent Statement:** Not applicable.

**Data Availability Statement:** Hydrometric data can be found in the SIR open repository at https://www.sir.toscana.it/consistenza-rete (accessed on 1 November 2021). Ecological monitoring data can be found in the ARPAT archive at https://sira.arpat.toscana.it/apex2/f?p=102:3:0 (accessed on 1 November 2021).

**Conflicts of Interest:** The authors declare no conflict of interest. The funders had no role in the design of the study.

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
