# Peer review of "WFD Ecological Quality Indicators Are Poorly Correlated with Water Levels in River Catchments in Tuscany (Italy)"

_hydrology, doi:10.3390/hydrology8040185_

Round 1

Reviewer 1 Report

The article is interesting and presents a straight forward analysis focusing on the relationship between ecological and hydrological indicators. The topic indeed attracts the interest of eco-hydrologists and hydrologists in general, as it is relevant with the implementation of the WFD and the concept of the e-flows. The manuscript is very well written, apart from some minor linguistic errors.

Overall, the article could be published in the Hydrology following a minor revision addressing a few issues mentioned below

  • Title: It is more accurate to say WFD ecological quality indicators and not WFD ecological status. The indicators that you describe define the ecological quality based on a specific BQE.
  • Line 12: I think here it would be useful to define the BQE for each indicator and also mention the name of the indices for macrophytes and diatoms for consistency.
  • Lines 56: Which indices? How are they called?
  • Line 69: I am a bit sceptical about the use of LIMeco as an ecological indicator. Since it describes the physicochemical conditions, which is a supportive element to the ecological quality. It is rather a chemical and not an ecological indicator. It is best to mention that in your article, perhaps in the methods. You could clarify that on top of the ecological indicators you used the LIMeco index as a proxy of the physicochemical conditions at the studied sites.
  • In table 1 do the values in colored cells mean that they are statistically significant? Did you examine that statistical significance of the correlations?
  • The discussion is well written. I recommend to further discuss about the biological indices and their potential to capture hydromorphological changes. Usually the macroinvertebrate indices are built to capture organic pollution because they can reflect the preferences of specific invertebrate families or guilds to certain levels and types of organic material. Instead, they do not use guilds that are representative of lotic or lentic conditions. What is the case with the STAR_ICMi index?
  • It would also be useful to mention a few words about the other indices as well (macrophytes and diatoms). Diatoms in general respond well to pollution but not to hydrological changes. Macrophytes might be better indicators of hydrology, depending the index. It is important to know a bit more about the indices.

Author Response

The article is interesting and presents a straight forward analysis focusing on the relationship between ecological and hydrological indicators. The topic indeed attracts the interest of eco-hydrologists and hydrologists in general, as it is relevant with the implementation of the WFD and the concept of the e-flows. The manuscript is very well written, apart from some minor linguistic errors. Overall, the article could be published in the Hydrology following a minor revision addressing a few issues mentioned below

General reply. Thank you for your comments which helped improving the clarity of the manuscript. Some minor grammar errors have been amended. You can find a point-by-point reply to the comments below.

  • Title: It is more accurate to say WFD ecological quality indicators and not WFD ecological status. The indicators that you describe define the ecological quality based on a specific BQE.

Reply. We agree, the title has been modified.

  • Line 12: I think here it would be useful to define the BQE for each indicator and also mention the name of the indices for macrophytes and diatoms for consistency.

Reply. We added to the abstract the acronyms of macrophytes Macrophyte Biological Index for Rivers (IBMR) and for diatom (TDI) Trophic Diatom Index and in the introduction the full name of the indices.

  • Lines 56: Which indices? How are they called?

      Reply. We added the axronym and extended name of the indices.in lines 56-57.

  • Line 69: I am a bit sceptical about the use of LIMeco as an ecological indicator. Since it describes the physicochemical conditions, which is a supportive element to the ecological quality. It is rather a chemical and not an ecological indicator. It is best to mention that in your article, perhaps in the methods. You could clarify that on top of the ecological indicators you used the LIMeco index as a proxy of the physicochemical conditions at the studied sites.

Reply. We agree that LIMeco is just a proxy of ecological quality because it accounts for dissolved oxygen, nitrogen and phosphorus, nevertheless it is widely used. We clarified this aspect in ll 54-56 where LIMeco is introduced.

  • In table 1 do the values in colored cells mean that they are statistically significant? Did you examine that statistical significance of the correlations?

Reply. Yes, the statistical significance of the correlations has been evaluated with t-test which rejected the null hypothesis at the 5% significance level. We clarified this aspect in ll 151-152.

  • The discussion is well written. I recommend to further discuss about the biological indices and their potential to capture hydromorphological changes. Usually the macroinvertebrate indices are built to capture organic pollution because they can reflect the preferences of specific invertebrate families or guilds to certain levels and types of organic material. Instead, they do not use guilds that are representative of lotic or lentic conditions. What is the case with the STAR_ICMi index?

Reply. The STAR_ICMI index is based on the assessment and normalization of different macro-invertebrates indicators which are mostly related to abundance, number of taxa and diversity of defined species. Thus, they do not specifically account for local hydraulic conditions and morphological characteristics. An example of indicator in this perspective is the Lentic-lotic River Descriptor (LRD) (Buffagni et al., 2009, https://doi.org/10.4081/jlimnol.2009.92 ) which was found significant in describing invertebrate communities. This aspect has been clarified in the discussion.

  • It would also be useful to mention a few words about the other indices as well (macrophytes and diatoms). Diatoms in general respond well to pollution but not to hydrological changes. Macrophytes might be better indicators of hydrology, depending the index. It is important to know a bit more about the indices.

Reply. In our study area, the diatom indicator (TDI) has been found one of the less sensitive in the monitored sites, while the IBMR and STAR_ICMi very sensitive ones, with the highest number of poor and very poor status. IBMR is based on field cover per-centage, the species trophic score, and a coefficient of ecological amplitude and has been found sensitive to discharge diversion in rivers also without pollution, in this sense the correlation (r=0.37) found with H1-H365, i.e, the water level range supports previous findings. This aspect has been clarified in the discussion around lines 192-198.

Reviewer 2 Report

The article is clear in its scientific reading and interpretation. The qualitative study is  presented in interesting approach linked to  the ecological status indicators. The objetive of this communication "... analyzing preliminarily the correlation between official ecological indicators ...and characteristics water levels.. for the river catchments in Tuscany" is too short face to the presented discussion.
The conceptual discussion should be more in-depth making the comparison with another studies in EU, namely to clarefy the obtained results in áreas with the same caracteristics.
The discussion   should be improved in order to present. in clear way, advantages, potentials and limitations of determination of ecological flows with a purely hydrological approach. It would interessant to introduce the climate change and the incertitude context for the sustain of the conclusions of the authors.

Author Response

The article is clear in its scientific reading and interpretation. The qualitative study is  presented in interesting approach linked to  the ecological status indicators. The objetive of this communication "... analyzing preliminarily the correlation between official ecological indicators ...and characteristics water levels.. for the river catchments in Tuscany" is too short face to the presented discussion.
The conceptual discussion should be more in-depth making the comparison with another studies in EU, namely to clarify the obtained results in áreas with the same caracteristics.
The discussion   should be improved in order to present. in clear way, advantages, potentials and limitations of determination of ecological flows with a purely hydrological approach. It would interessant to introduce the climate change and the incertitude context for the sustain of the conclusions of the authors.

Reply. Thank you for your comments. The discussion has been modified according to your suggestions and the two other referees’ comments and includes the following: (1) a comparison with other studies which observed similar poor correlation between ecological and hydrological indicators; (2) issues related to the capability of official ecological indicators to fully detect hydrological alterations; (3) limitations of the observations used in the study area; (4) advantages of hydrological approached in e-flows determination in terms of costs, time and area of application; (5) predicted and observed trends in low flows which could potentially affect both ecological and hydrological variables; (6) future research directions; (7) need for a denser observation network of both ecological and hydrological indicators for waters resources planning in the context of climate change.

Reviewer 3 Report

  1. The manuscript presents WFD ecological status indicators are poorly correlated with water levels in river catchments in Tuscany (Italy), which is interesting. The subject addressed is within the scope of the journal.
  2. However, the manuscript, in its present form, contains several weaknesses. Appropriate revisions to the following points should be undertaken in order to justify recommendation for publication.
  3. Full names should be shown for all abbreviations in their first occurrence in texts. For example, STAR-ICMi in p.1, LIMeco a in p.1, EU a in p.2, GIS a in p.3, etc.
  4. For readers to quickly catch your contribution, it would be better to highlight major difficulties and challenges, and your original achievements to overcome them, in a clearer way in abstract and introduction.
  5. 1 - STAR-ICMi, LIMeco, macrophytes and diatom are adopted as ecological indicators in this study. What are the other feasible alternatives? What are the advantages of adopting these indicators over others in this case? How will this affect the results? More details should be furnished.
  6. 2 - Tuscany is adopted as the case study. What are other feasible alternatives? What are the advantages of adopting this case study over others in this case? How will this affect the results? The authors should provide more details on this.
  7. 3 - stage-duration curve is adopted for water level. What are other feasible alternatives? What are the advantages of adopting this approach over others in this case? How will this affect the results? The authors should provide more details on this.
  8. 3 - six reference water levels are adopted for correlation analysis. What are the other feasible alternatives? What are the advantages of adopting these water levels over others in this case? How will this affect the results? More details should be furnished.
  9. 3 - a geospatial association in a GIS environment is adopted to combine hydrological and ecological status datasets. What are other feasible alternatives? What are the advantages of adopting this approach over others in this case? How will this affect the results? The authors should provide more details on this.
  10. 4 - Pearson's correlation coefficient is adopted for correlation analysis. What are other feasible alternatives? What are the advantages of adopting this method over others in this case? How will this affect the results? The authors should provide more details on this.
  11. 6 - “…the LIMeco index, which is more correlated to characteristic water levels, can be seen as a proxy of the river capability to dilute nutrients and organic load. The LIMeco index is nevertheless one of the less sensitive indicators in the study area according to ARPAT reports [36].…” Some justification should be furnished on this issue.
  12. Some key model parameters are not mentioned. The rationale on the choice of the set of parameters should be explained with more details. Have the authors experimented with other sets of values? What are the sensitivities of these parameters on the results?
  13. Some assumptions are stated in various sections. Justifications should be provided on these assumptions. Evaluation on how they will affect the results should be made.
  14. The discussion section in the present form is relatively weak and should be strengthened with more details and justifications.
  15. Moreover, the manuscript could be substantially improved by relying and citing more on recent literatures about contemporary real-life case studies of sustainability and/or uncertainty such as the followings. Discussions about result comparison and/or incorporation of those concepts in your works are encouraged:
  • Zhou, Y.L., et al., “Real-Time Probabilistic Forecasting of River Water Quality under Data Missing Situation: Deep Learning plus Post-Processing Techniques,” Journal of Hydrology 589: 125164 2020.
  • Shamshirband, S., et al., “Ensemble models with uncertainty analysis for multi-day ahead forecasting of chlorophyll a concentration in coastal waters,” Engineering Applications of Computational Fluid Mechanics 13 (1): 91-101 2019.
  • Tiyasha, et al., “A survey on river water quality modelling using artificial intelligence models: 2000-2020,” Journal of Hydrology 585: 124670 2020.
  1. The conclusion section is missing. In the conclusion section, the limitations of this study, suggested improvements of this work and future directions should be highlighted.

Author Response

The manuscript presents WFD ecological status indicators are poorly correlated with water levels in river catchments in Tuscany (Italy), which is interesting. The subject addressed is within the scope of the journal.

However, the manuscript, in its present form, contains several weaknesses. Appropriate revisions to the following points should be undertaken in order to justify recommendation for publication.

General reply. Thank you for your review, you can find below a point by point reply to all the comments raised.

  1. Full names should be shown for all abbreviations in their first occurrence in texts. For example, STAR-ICMi in p.1, LIMeco a in p.1, EU a in p.2, GIS a in p.3, etc.

Reply. the description of the abbreviation has been given at the first occurrence in text.

  1. For readers to quickly catch your contribution, it would be better to highlight major difficulties and challenges, and your original achievements to overcome them, in a clearer way in abstract and introduction.

Reply. The main difficulties in the understanding of the link between ecological and hydrological indicators have been better highlighted in the abstract and at the end of the introduction.

  1. STAR-ICMi, LIMeco, macrophytes and diatom are adopted as ecological indicators in this study. What are the other feasible alternatives? What are the advantages of adopting these indicators over others in this case? How will this affect the results? More details should be furnished.

Reply. The adopted indicators are the result of some years of debate of the European scientific community to identify a limited number of significant parameters sensitive to pollution and anthropic pressures on surface water bodies. They are in use (officially) since 2010 in Italy for the requirements of the European Directive 60/2000/EC (WFD) thus it is not possible to determine how the use of other indicators could affect the results. It is in the same framework (WFD) that ecological flows should be determined so this study is consistent with the current practice in the study area.

  1. Tuscany is adopted as the case study. What are other feasible alternatives? What are the advantages of adopting this case study over others in this case? How will this affect the results? The authors should provide more details on this.

Reply. Tuscany is adopted as a case study because among the authors of the manuscript there are representatives of the River District Authority which is in charge of defining ecological flows for their competent river network (in Tuscany). Clearly the results are specific for the study area, as the title clearly recognizes, however similar results have been found in other Italian studies in different geographic context as specified in the text LL. 47-55 and LL. 183-186, 191-193 References [5, 18-23].

  1. stage-duration curve is adopted for water level and six reference water levels are adopted for correlation analysis. What are other feasible alternatives? What are the advantages of adopting this approach over others in this case? How will this affect the results? The authors should provide more details on this.

Reply. Stage-duration curves and discharge-duration curves are a common tool in hydrology to identify the permanence of assigned values as described in ref [38-39-40] and were adopted to identify minimum vital flows in natural conditions. Another feasible alternative would be to use the indicators of hydrologic alteration (IHA) developed by Richter et al (1997) [12] which however is more suitable to determine hydrological changes before and after anthropic interventions, e.g., hydropower plant construction. Nevertheless, some of the IHA parameter coincide with those adopted in this study (in terms of water levels), i.e., the minimum flow (H365), maximum flow (H1), maximum-minimum flow (H1-H365). This aspect has been clarified in the text.

  1. a geospatial association in a GIS environment is adopted to combine hydrological and ecological status datasets. What are other feasible alternatives? What are the advantages of adopting this approach over others in this case? How will this affect the results? The authors should provide more details on this.

Reply. When the ecological and hydrological datasets are not already available in combination, we believe there aren’t other alternatives to pair the data. The possible uncertainties of this approach are mentioned around line 216 in the discussion section.

  1. Pearson's correlation coefficient is adopted for correlation analysis. What are other feasible alternatives? What are the advantages of adopting this method over others in this case? How will this affect the results? The authors should provide more details on this.

Reply. Pearson’s correlation coefficient is widely adopted in statistics for measuring linear correlation between two sets of data and colloquially is known as “the correlation coefficient” it should be thus well known by most of the readers without specific justifications.

  1. “…the LIMeco index, which is more correlated to characteristic water levels, can be seen as a proxy of the river capability to dilute nutrients and organic load. The LIMeco index is nevertheless one of the less sensitive indicators in the study area according to ARPAT reports [36].…” Some justification should be furnished on this issue.

Reply. We clarified this aspect in the text as follows “the LIMeco index, which is more correlated to characteristic water levels, can be seen as a proxy of the river capability to dilute nutrients and organic load, it is indeed more a chemical indicator sensitive to river discharges than an ecological indicator. The LIMeco index is nevertheless one of the less sensitive indicators in the study area according to ARPAT reports and should not be considered alone as a reference for ecological flows determination”

  1. Some key model parameters are not mentioned. The rationale on the choice of the set of parameters should be explained with more details. Have the authors experimented with other sets of values? What are the sensitivities of these parameters on the results?

Reply. There isn’t any model applied in this work, consequently there aren’t model parameters, only statistical analysis of the frequencies of water levels (stage-duration curve) and correlation analysis of ecological and hydrological indicators.

  1. The discussion section in the present form is relatively weak and should be strengthened with more details and justifications. Moreover, the manuscript could be substantially improved by relying and citing more on recent literatures about contemporary real-life case studies of sustainability and/or uncertainty such as the followings. Discussions about result comparison and/or incorporation of those concepts in your works are encouraged:
  • Zhou, Y.L., et al., “Real-Time Probabilistic Forecasting of River Water Quality under Data Missing Situation: Deep Learning plus Post-Processing Techniques,” Journal of Hydrology 589: 125164 2020.
  • Shamshirband, S., et al., “Ensemble models with uncertainty analysis for multi-day ahead forecasting of chlorophyll concentration in coastal waters,” Engineering Applications of Computational Fluid Mechanics 13 (1): 91-101 2019.
  • Tiyasha, et al., “A survey on river water quality modelling using artificial intelligence models: 2000-2020,” Journal of Hydrology 585: 124670 2020.

Reply. Thank you for suggesting this interesting works which have been incorporated in the discussion as successful example of AI applied to water quality modelling.

  1. The conclusion section is missing. In the conclusion section, the limitations of this study, suggested improvements of this work and future directions should be highlighted.

Reply. The conclusion section is not mandatory according to the Journal guidelines and the manuscript organization is free for communications. The limitations of the study and future research directions are highlighted in the discussion.

Round 2

Reviewer 3 Report

The revised paper has addressed all my previous comments, and I suggest to ACCEPT the paper as it is now.

Author Response

Thank you for your revision. We went again through the manuscript to correct typos and spell check.

best regards